# Computer-assisted analysis of polysomnographic recordings improves inter-scorer associated agreement and scoring times

Diego Alvarez-Estevez[1]*, Roselyne M. Rijsman[2]

**1** Center for Information and Communications Technology Research (CITIC), Universidade da Coruña, A Coruña, Spain, **2** Sleep Center and Clinical Neurophysiology Department, Haaglanden Medisch Centrum, The Hague, The Netherlands

* diego.alvareze@udc.es

**Data Availability Statement:** All relevant data are within the paper and its Supporting Information files.

## Abstract

### Study objectives

To investigate inter-scorer agreement and scoring time differences associated with visual and computer-assisted analysis of polysomnographic (PSG) recordings.

### Methods

A group of 12 expert scorers reviewed 5 PSGs that were independently selected in the context of each of the following tasks: (i) sleep staging, (ii) scoring of leg movements, (iii) detection of respiratory (apneic-related) events, and (iv) of electroencephalographic (EEG) arousals. All scorers independently reviewed the same recordings, hence resulting in 20 scoring exercises per scorer from an equal amount of different subjects. The procedure was repeated, separately, using the classical visual manual approach and a computer-assisted (semi-automatic) procedure. Resulting inter-scorer agreement and scoring times were examined and compared among the two methods.

### Results

Computer-assisted sleep scoring showed a consistent and statistically relevant effect toward less time required for the completion of each of the PSG scoring tasks. Gain factors ranged from 1.26 (EEG arousals) to 2.41 (leg movements). Inter-scorer kappa agreement was also consistently increased with the use of supervised semi-automatic scoring. Specifically, agreement increased from = 0.76 to K = 0.80 (sleep stages), = 0.72 to K = 0.91 (leg movements), = 0.55 to K = 0.66 (respiratory events), and = 0.58 to = 0.65 (EEG arousals). Inter-scorer agreement on the examined set of diagnostic indices did also show a trend toward higher Interclass Correlation Coefficient scores when using the semi-automatic scoring approach.

**Funding:** This study has been initiated at Haaglanden Medisch Centrum with a cost neutral policy under project number 2019-073. One of the authors (DAE) has received funding under project ED431H 2020/10 of Xunta de Galicia. Authors wish to acknowledge the support received from the Centro de Investigación de Galicia (CITIC), funded by Xunta de Galicia and the European Union (European Regional Development Fund-Galicia 2014-2020 Program), by grant ED431G 2019/01. The funders had no role in study design, data collection and analysis, decision to publish, or preparation of the manuscript.

**Competing interests:** The authors have declared that no competing interests exist.

## Conclusions

Computer-assisted analysis can improve inter-scorer agreement and scoring times associated with the review of PSG studies resulting in higher efficiency and overall quality in the diagnosis sleep disorders.

## Introduction

The analysis of polysomnographic sleep recordings (PSGs) constitutes one of the most time-consuming tasks in the daily work of a Sleep Center. A typical PSG examination contains somewhere between eight up to twenty-four hours of continuous neurophysiological activity recording. Common PSG data include, among others, different traces of electroencephalographic (EEG), electrooculographic (EOG), electrocardiographic (ECG), electromyographic (EMG), and respiratory activity [1]. Likewise, analysis of the PSG can be organized into different subtasks, for instance, analysis of the macro and micro structure of sleep, characterization of the respiratory function, or the scoring of limb movement activity.

Clinical findings over the last years have uncovered the negative consequences that Sleep Disorders exert over health, contributing to the general public awareness. This situation has led to a steady increase in the demand for PSG investigations, which represents a challenge for the already congested sleep centers. Clinician's time is expensive and scant. In addition, the large amount and the complexity of the associated data, makes of PSG analysis a task prone to errors and to subjective interpretations [2]. Indeed, despite homogenization procedures promoted by development and usage of clinical standard guidelines [1,3], different grades of intra- and inter-expert variability have been reported in the literature, affecting the resulting PSG outcomes, which vary among the specific references or tasks subject to evaluation [4–9]. In this context, the use of automatic scoring algorithms presents potential advantages. First, as computer analyses produce deterministic (repeatable) outputs, they have the capacity to overcome the variability associated with intrinsic human subjectivity, thereby contributing to standardization of the process and overall quality improvement. On the other hand, automatic scoring would result in great savings in terms of scoring time, hence human resources, reducing overall costs of the diagnosis. Literature, in fact, is rich on examples that focus on the development and validation of automatic analysis methods in different areas related to the scoring of sleep studies [5,10–19]. However, despite recent advances and the promising man-machine agreement results reported in some of these works, reliance on automatic scoring among the clinical community remains low [15,20,21]. Moreover, there are still open questions on whether "are we there yet", in terms of acceptable performance and enough generalization capabilities of these algorithms as compared to well-trained human clinicians [2].

An alternative approach is the so-called "semi-automatic" scoring, whereby an automatic algorithm performs a preliminary first analysis pass, after which the results are reviewed by an expert clinician who corrects possible miss-scorings produced by the computer program. Because of the expert supervision, the procedure can still be safely implemented in the clinical practice: would the (full) automatic scoring algorithm make a mistake (e.g. due to data of varying quality and/or different patient phenotypes) the error can still be detected and corrected by the supervising expert. An open question remains, though, as to whether economics of semi-automatic scoring still apply, in particular, because of keeping expert intervention as part of the process, and evaluation of its possible role in reducing the baseline levels of manual inter-scorer variability.

The main goal of this study is to evaluate the possible benefits of using semi-automatic analysis for the scoring of PSGs in comparison to baseline levels of human performance. Performance, in the context of this study, is characterized by the use of objective (quantifiable) metrics regarding the respective scoring times and resulting levels of inter-rater scoring variability. Specifically, in this study, experiments are carried out, independently and systematically, in the context of the following four (usually, the most important) PSG analysis subtasks: sleep staging, scoring of leg movements, and detection of respiratory (apneic-related) events and of EEG arousals.

Our work is of interest, as the topic has barely been examined in the available literature, with perhaps some few exceptions regarding the specific subtask of sleep staging [10,20,22,23]. Regardless, to our knowledge, no previous studies have attempted to examine the hypothesis on whether semi-automatic scoring can contribute to reduction of inter-scorer variability in a systematic way. Likewise, we believe this is the first study to systematically address possible scoring time differences between the manual and semi-automatic approaches.

To have objective (measurable) references of the levels of scoring performance is as well of fundamental importance to allow implementation of quality control mechanisms in the patient care. Our work contributes as well by adding to the existing literature on manual scoring, taking into account that evolution of the scoring methods and reference clinical guidelines motivates reassessment of the existing references, for which some of them might be outdated. Furthermore, for some of the examined scoring tasks, literature references of related quality metrics examined in this work have never been reported before.

## Methods

### Study database

PSG data for this study has been gathered by retrospective inspection of the Haaglanden Medisch Centrum (HMC, The Hague, The Netherlands) Sleep Center patient database. The pre-sample dataset comprised 2801 recordings, corresponding to the most recent full-year data in the HMC database at the time (2019). The final inclusion dataset, involving 20 PSG recordings, was selected from this initial sample using an automatic selection procedure implemented with the aim to minimize the risk of selection bias. The automatic selection procedure is described in further detail in the later section "Selection of PSGs".

All data were originally acquired in the course of common clinical practice. PSG data consisted of raw biomedical signals following standard acquisition procedures described in the AASM guidelines [1]. SOMNOscreen$^{TM}$ plus devices (SOMNOmedics, Germany) were used as the acquisition hardware. Scoring annotations resulting from regular clinical examination workflow accompanied each recording. Clinical scorings were carried out by HMC expert sleep technicians including the analysis of sleep stages, EEG arousals, and detection of respiratory events following the standard AASM guidelines [1], and scoring of leg movements according to the WASM2016 manual [3]. Both the raw signals' data and the resulting clinical scoring annotations were digitally stored using the EDF+ format [24].

All recordings were de-identified and subrogate study numbers were assigned to each patient prior their inclusion in this study, thus avoiding any possibility of individual patient identification. Under these conditions the study obtained approval of the local Medical Ethics Committee (Medisch Ethische Toestsingscomissie Zuidwest Holland) under code MTEC-19-065, who considered that the protocol did not fall under the scope of the Medical Scientific Research Involving Human Subjects Act (WMO) and that no explicit informed consent was required by participants. Study has as well obtained written permission from the database owner for publication.

## Rescoring task

In the present study a group of 12 expert scorers were prompted to review 5 PSGs that were independently selected in the context of each of the following tasks: (i) sleep staging, (ii) scoring of leg movements, (iii) detection of respiratory events, and (iv) detection of EEG arousals. All scorers were experienced sleep technicians from the same center (HMC), who have a completed training certification, and that regularly and autonomously participate in the daily scoring routine of the sleep department. Sleep technicians with uncompleted training or undergoing supervision were excluded from this study.

Rescoring was repeated, separately for each task, using first a purely manual (visual), followed by a semi-automatic scoring approach. A total of 20 different PSG recordings were included in the final study dataset, hence resulting in a total 40 different scoring exercises per scorer. Each participant scorer was tasked to review the exact same recordings, and on each case scoring was limited to the specific task under consideration. In all cases, scoring was blindly performed to both the patient identity (by using de-identified recordings) and to the results of possible previous scorings (e.g. that could take place during regular clinical workflow, from other scorers, or during a previous self-rescoring subtask).

To avoid learning effects, at least 4 months of separation between these two manual and semi-automatic scoring moments were scheduled. For reference, an average amount of 70 PSG recordings are scored by each sleep technician due to the normal sleep lab activity during that period. Scorers were also not informed about the fact that manual and semi-automatic scorings would involve the exact same recordings.

All scoring tasks took place using the Polyman software [25]. For each task, a timer was automatically set in the background by the program (unavailable to the human scorer). The tick counting was automatically paused if no mouse or keyboard interaction was detected during more than a minute, and the offline time was subtracted from the total scoring time. The resulting active scoring time periods were saved separately in a file for later analysis.

Scoring took place between Time In Bed (TIB) periods only (between "lights off" and "lights on" markers), which were provided as pre-filled annotations. For the scoring of leg movements, respiratory events, and of EEG arousals, the pre-filled clinical hypnogram was also provided as additional source for contextual interpretation and to avoid divergence of initial conditions. Scorers were instructed to stick to the scoring of the relevant events in the context of the specific target task, not being allowed to change any pre-filled contextual information, when supplied.

For implementation of the semi-automatic scoring process, the annotations that resulted from the output of the corresponding automatic analysis algorithms were provided, in addition, at the start of the scoring. Scorers were instructed to review these scorings by adding, deleting, or editing the event's onset and offset times, where corresponds, and according to their own expertise. Details regarding the development and validation of the automatic scoring algorithms that were used for this purpose have been reported in past works. The reader is referred to check the corresponding references regarding the automatic scoring of sleep stages [26], leg movement activity [27,28], respiratory events [29,30], and EEG arousals [31,32]. A free-version of the Polyman software and source code for non-licensed versions of automatic scoring are also accessible online [33].

## Selection of PSGs

For each of the target scoring tasks (i.e. sleep staging, leg movements, respiratory events, and EEG arousals) 5 PSGs were independently and automatically selected from the initial pre-sample dataset. Sampling size was determined by the limited time availability of the expert scorers

allocated for the study. Under these circumstances, independent per-task selection was preferred, rather than recording-level selection, with the aim to obtain the best-fit representatives for each task, avoiding unnecessary within-subject dependencies for sticking with the same PSG for all the four scoring tasks (i.e. the same PSG, for instance, might represent an interesting sleep staging analysis scenario, but contain irrelevant leg movement or EEG arousal scoring cases).

With that in mind, an automatic selection procedure was implemented with the objective to minimize the chance of selection bias and obtain a balanced representation of scoring difficulty for each task. The underlying hypothesis correlates scoring difficulty with scoring time and inter-scorer variability: the more difficult a PSG becomes for manual scoring, the more time it would take and the more inter-scorer variability would be associated to its scoring, and viceversa. Regardless, within the implemented procedure no specific exclusion criteria were applied to filter out recordings due to specific patient conditions, or poor signal quality. A sufficient condition was that the recording had been accepted for manual scoring during regular clinical workflow, a condition that, by definition, was already satisfied by all recordings included in the pre-sample dataset. The underlying motivation was to reproduce, as close as possible, the same conditions as in real clinical practice and consider the most complete representation of the general patient phenotype.

Hence, for each of the four scoring subtasks, the following selection procedure was scheduled using the human-automatic agreement as subrogate of the associated scoring difficulty:

i. First, taking as reference the complete pre-sample dataset (2801 PSGs) full automatic analysis (no human intervention at all) of each recording was performed. This analysis led to a list of automatically scored events $L_a(i)$, for each recording $i$, related to the corresponding scoring task under consideration.

ii. Using the list of automatically-generated events, $L_a(i)$, each PSG was compared with the corresponding list of events that resulted from clinical manual examination, $L_c(i)$. Confronting $L_a(i)$ with $L_c(i)$, a preliminary metric of performance agreement between the two scoring outputs, $K_{ac}(i)$, was obtained. Specifically, $K_{ac}$ was calculated using the Cohen´s Kappa statistic [34]. Details on the implementation of $K_{ac}$ for each of the four target subtasks are described in the section "analysis methods".

iii. By repeating this operation through all 2801 PSGs available in the initial pre-sample dataset, a distribution $DK_{ac}$ of $K_{ac}(i)$ values was obtained.

iv. Using $DK_{ac}$ as reference, uniform sampling was performed to select the target number ($n = 5$) of recordings to be included in each subtasks' final study dataset. Specifically, the 5 recordings whose associated $K_{ac}(i)$ performance metrics represent the middle of each inter-quartile range, plus the median, were selected as representatives of their respective populations. In other words, the recordings with performance scores representing the 12.5th, 37.5th, 50th, 62.5th and 87.5th percentiles of each $DK_{ac}$ distribution were selected for the final study dataset.

Effectively, the above described procedure is preferable over random resampling as it avoids potential selection of outliers by chance (i.e. extreme favorable or unfavorable cases for the automatic algorithm) that might bias the resulting sample. Similar selection procedures were scheduled during the validation of different automatic scoring algorithms that were reported in the past [26,32]. Correlation analyses for the validation of the selection hypothesis are provided and discussed in S4 Appendix.

**Table 1. Summary of general demographics and PSG descriptors in the study dataset.**

| Parameter | Task group | | | | |
|---|---|---|---|---|---|
| | Sleep staging | Leg movements | Respiratory events | EEG arousals | All |
| n | 5 | 5 | 5 | 5 | 20 |
| Age (years) | 52.0 [47.0, 57.0] | 57.0 [51.0, 68.0] | 59.0 [57.0, 61.0] | 55.0 [52.0, 63.0] | 57.0 [51.8, 61.5] |
| Male (n, %) | 5 (100%) | 1 (20%) | 3 (60%) | 3 (60%) | 12 (60%) |
| Time In Bed (TIB, hours) | 7.5 [7.4, 8.0] | 7.3 [7.0, 7.3] | 6.5 [6.4, 7.4] | 8.1 [7.2, 8.2] | 7.3 [7.0, 8.0] |
| Total Sleep Time (TST, hours) | 5.9 [5.9, 7.1] | 6.7 [5.9, 6.7] | 6.0 [5.8, 6.1] | 7.3 [5.5, 7.4] | 6.0 [5.7, 7.0] |
| Sleep Latency (SL, min) | 4.6 [3.0, 5.4] | 1.8 [1.5, 3.3] | 9.7 [2.6, 14.5] | 1.9 [1.0, 21.5] | 3.1 [1.6, 16.2] |
| Stage R latency (min) | 69.5 [40.0, 174.0] | 104 [64.5, 124.0] | 59.0 [52.0, 138.0] | 83.5 [80.0, 85.0] | 81.8 [58.0, 127.0] |
| Wake After Sleep Onset (WASO, min) | 72.8 [24.7, 124.0] | 53.0 [39.9, 68.2] | 38.9 [28.1, 52.2] | 89.6 [46.7, 100.4] | 52.6 [27.3, 106.3] |
| Sleep Efficiency (SE, %) | 91.1 [74.2, 94.5] | 88.6 [83.8, 90.9] | 89.9 [84.7, 92.8] | 83.0 [76.6, 90.5] | 89.2 [76.0, 93.2] |
| Arousal Index (ArI, n/TST) | 4.7 [1.8, 17.6] | 14.3 [13.4, 16.0] | 19.8 [9.6, 23.4] | 9.6 [7.9, 13.7] | 13.5 [7.5, 20.7] |
| Apnea-Hypopnea Index (AHI, n/TST) | 9.1 [6.0, 9.4] | 5.1 [0.6, 20.7] | 6.2 [5.8, 13.1] | 7.2 [3.0, 20.6] | 6.7 [5.0, 20.6] |
| Oxygen Desaturation Index (ODI, n/TST) | 10.7 [10.5, 10.8] | 1.4 [0.6, 24.8] | 9.9 [5.5, 15.4] | 9.9 [2.8, 15.7] | 10.2 [4.4, 18.0] |
| Leg Movement Index (n/TST) | 52.3 [9.0, 56.5] | 32.0 [21.0, 57.7] | 14.2 [8.6, 39.8] | 30.0 [19.3, 43.2] | 31.0 [12.9, 53.4] |
| Periodic Leg Movement Index (PLMI, n/TST) | 13.0 [0.3, 45.7] | 13.8 [13.4, 49.0] | 0.8 [0.4, 26.7] | 22.4 [4.1, 28.4] | 13.6 [0.7, 32.8] |

PSG descriptors correspond to values resulting from retrospective examination in the clinical database, i.e. prior to the multi-expert rescoring procedures carried out in this study. Distributions are characterized using the median and the corresponding interquartile ranges.

Table 1 summarizes the general demographics and PSG descriptors of the resulting patient study sample. Data are presented stratified among the corresponding task-specific subgroups.

## Analysis methods

Analysis of inter-scorer agreement is carried out in the first place by discretizing the recording time into non-overlapping analysis mini-epochs. Each analysis mini-epoch is assigned the corresponding scorer's output in the context of the specific target subtask. Duration of the mini-epochs are task-related as well. In the case of the sleep scoring, analysis epochs have the standard duration of 30s and take possible values according to the AASM clinical guidelines, that is, either W, N1, N2, N3, or R [1]. In the context of the leg movements, respiratory events, and EEG arousals' scoring subtasks, each mini-epoch takes a binary value noting the presence or absence of event, respectively, if overlapping or not with the events marked by the scorer. Analysis mini-epoch duration is set to 0.5s for all the three subtasks.

Time discretization in the above terms leads to the construction of $k$-dimensional contingency tables ($k = 5$ for sleep staging, $k = 2$ otherwise) from which standard metrics of agreement for categorical data can be derived. Within each task, agreement between each of the twelve scorers' pair combination (n = 66 per recording) is calculated using the Cohen's kappa statistic. The use of the Cohen's kappa is motivated given its widespread use in the field, as well as its robustness in the case of imbalanced class distributions as it corrects for agreement due to chance [34].

Inter-scorer agreement is also evaluated among the diagnostic indices resulting from the respective scorings. Following the list of recommended parameters to be reported in PSG studies [1,3], a representative subset for each of the subtasks targeted in this study is selected. In particular, sleep quality-related parameters of Sleep Efficiency (SE), Sleep Onset Latency (SOL), and Wake After Sleep Onset (WASO) [35], in relation to the sleep scoring task; Apnea-Hypopnea Index (AHI), Apnea Index (AI), Hypopnea Index (HI) and Oxygen Desaturation Index (ODI), in relation to the scoring of respiratory events; Arousal Index (ArI), in relation

to the scoring of EEG arousals; and Leg Movement Index (LMI) and the Periodic Leg Movement Index (PLMI), in relation to the leg movements' scoring task. LMI and PLMI indices are calculated according to the WASM2016 scoring guidelines, the former being defined as the number of leg movements $\geq 0.5s$ after bilateral combinations per hour of sleep, with the latter including respiratory-related LMs as well in the counts [3]. Inter-scorer agreement among the resulting indices on each case (n = 12 per recording) is evaluated using the Intraclass Correlation Coefficient (ICC) [36]. Specifically, a two-way absolute single-measures variant of the statistic, ICC(A,1), is used [37]. A Matlab implementation for calculation of this coefficient has been used whose source code is available at [38].

Hypothesis testing is carried out to check for significant differences between the manual and semi-automatic scoring approaches. For this purpose, the reference level for statistical significance is set to $\alpha = 0.05$. Differences are examined using the paired version of the Wilcoxon signed rank test among all the matched kappa scorer pair combinations (n = 66 per recording, n = 330 in total for each task). Analogous analysis is performed for checking out differences in the respective scoring times among the matched individual scorers (n = 12 per recording, n = 60 in total for each task). For each test the corresponding effect size is reported using the Cohen's *D* statistic. Statistical significance on inter-scorer ICC agreement differences among diagnostic indices is also evaluated. For this purpose, the a priori expected agreement (*r0*) for the semi-automatic approach is set to match the effective ICC levels achieved with manual scoring.

Results of the above-mentioned analyses are presented in the subsequent section by aggregating the respective scorings among the five recordings involved within each scoring task. In order to keep the main text extension attainable, individualized per-recording results are provided as Supplementary Information (S1–S3 Appendixes). In this case, manual vs. semi-automatic differences in diagnostic indices are examined, again, using paired analyses. Comparison of the respective variance distributions is examined using the Brown-Forsythe (unpaired) test. For the latter, i.e. comparison of distribution's variance, the corresponding manual and semi-automatic indices are first mean normalized within their respective distributions to avoid possible bias due to differences in the respective population means.

## Results

### Analysis of scoring time

Fig 1 shows the median scoring time associated with the completion of the different analysis tasks according to the followed approach, i.e. manual or semi-automatic. Values on the bar plot are shown in minutes and aggregate the results among the five recordings involved on each case.

Table 2 expands the results of Fig 1 and shows the results of the associated statistical analyses involving the two scoring approaches. Data in Table 2 unveil a consistent and statistically relevant effect toward less time required for the completion of each task when using the semi-automatic scoring approach. Gain factors vary per task, with the largest time savings relating to the scoring of leg movements, followed by the analysis of the respiratory activity, and a less pronounced effect associated with the sleep staging task and the scoring of EEG arousals. The associated effect sizes on each case support these interpretations. In this regard, notice that a positive sign on the corresponding index indicates that the overall effect (in this case scoring time) is bigger in the manual scoring scenario, with the associated absolute value being an indicative of how much bigger the effect is.

When comparing absolute time values among the different tasks, our results show that detection of leg movements is the most time-consuming task when using manual analysis.

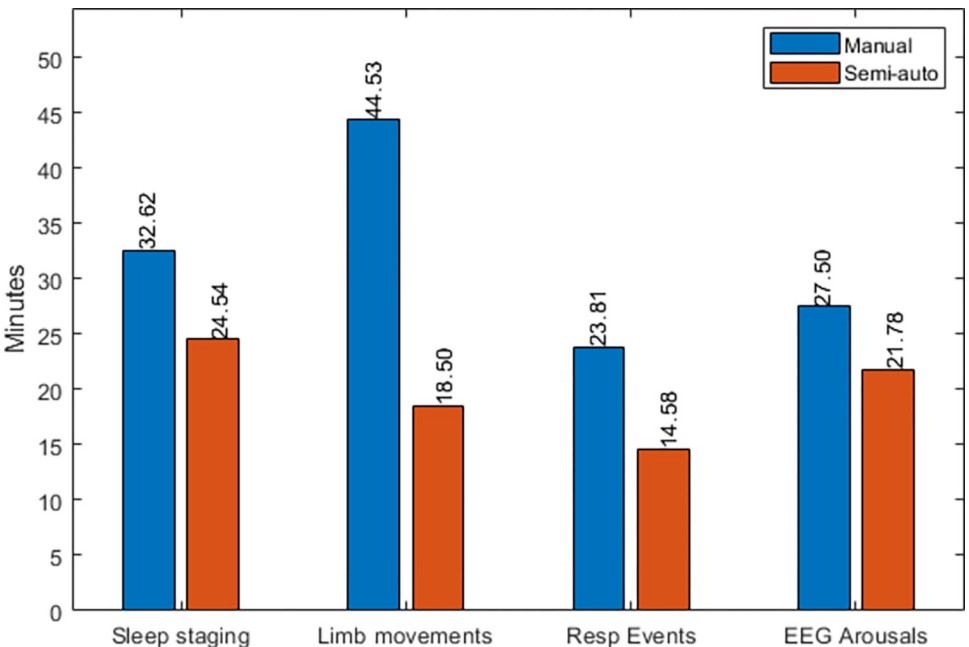

**Fig 1. Differences in scoring time between manual and semi-automatic scoring approaches.** Median scoring time per task is shown in minutes.

Scoring of respiratory events is relatively the quickest. The trend changes a bit when using the semi-automatic approach, resulting in sleep staging being the slowest, with analysis of respiratory activity remains as the fastest task.

Individualized per-recording and per-scorer analyses for each task can be found, respectively, in Tables A1-A4 and Figs A1-A4 in S1 Appendix.

## Analysis of kappa agreement

Fig 2 shows the global kappa agreement results per scoring task when comparing manual and semi-automatic scoring approaches. Values on the bar plot represent the median expert paired agreements among the five recordings within the corresponding task.

Table 3 shows results of the statistical analyses between the corresponding manual and semi-automatic scoring differences. Moreover, results are subcategorized for some of the tasks into different contexts of clinical interest. In particular, differences between wake and sleep

**Table 2. Analysis of scoring time differences per task between manual and semi-automatic approaches.**

| Scoring task | Manual | Semi-Auto | Gain factor | *p*-value | Effect size |
|---|---|---|---|---|---|
| Sleep staging | 32.62 [21.74, 48.64] | 24.54 [16.25, 39.81] | 1.33 | 0.0005* | 0.4297 |
| Leg movements | 44.53 [31.00, 65.30] | 18.50 [12.63, 26.73] | 2.41 | < 0.0001* | 1.3475 |
| Respiratory events | 23.81 [17.62, 46.72] | 14.58 [10.46, 20.68] | 1.63 | < 0.0001* | 0.9474 |
| EEG arousals | 27.50 [21.22, 37.65] | 21.78 [15.96, 28.59] | 1.26 | 0.0011* | 0.4233 |
| Altogether | 134.92 [113.08, 187.65] | 80.59 [66.75, 107.89] | 1.67 | < 0.0001* | 1.7527 |

n = 60 resulting from all twelve scoring experts and the five analyzed recordings for each of the corresponding tasks. Distributions are characterized using the corresponding median and interquartile ranges in minutes. Gain factors are calculated on each case as the ratio between the corresponding median scoring times.
*Statistically significant result.

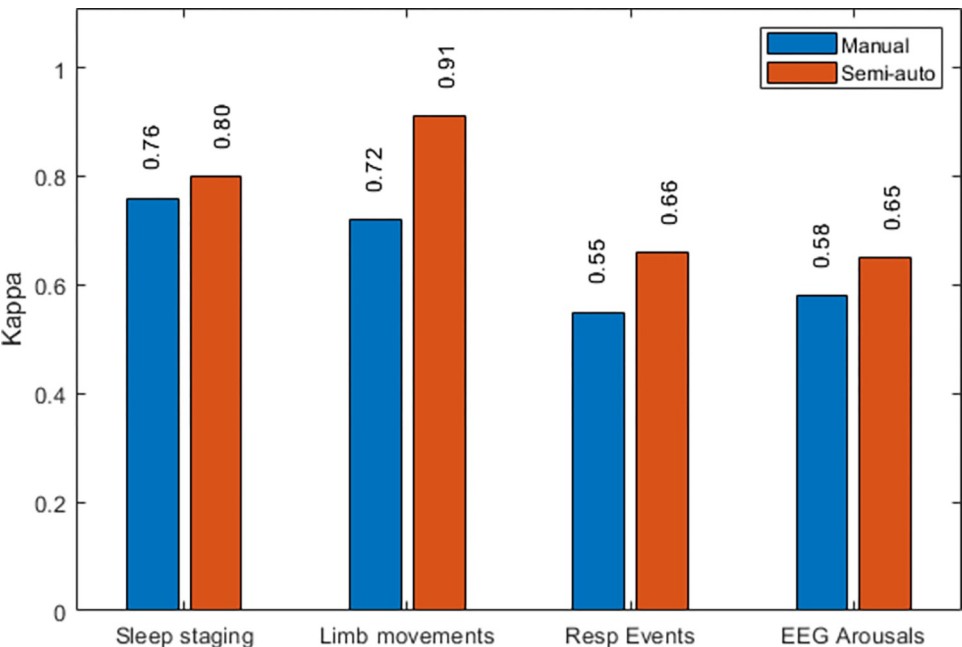

**Fig 2. Agreement comparison between manual and semi-automatic scoring approaches.** Median Kappa agreement for each task is shown by aggregating all the expert pair combinations throughout the five corresponding recordings.

periods are reported for leg movements, as well as for different types of respiratory (apneic-related) events. For the analysis of leg movements, the individual kappa scores for each individual channel (left / right leg) were averaged together before statistical analysis was executed.

Results from Table 3 show that statistically significant differences between manual and semi-automatic scoring are reached regardless of the specific task or the event subtype. A consistent trend toward higher inter-scorer agreement associated with the use of semi-automatic scoring is shown. Notice the associated effect sizes overall show a negative sign, being indicative of the general smaller agreement achieved in the manual scoring scenario. The highest absolute effect in this context is associated with the leg movements' detection task.

When comparing among the different tasks, the highest (either manual or semi-automatic) agreements are achieved in the case of the sleep staging and leg movements' detection tasks.

**Table 3. Overall kappa inter-scorer agreement per scoring task and comparison between manual and semi-automatic approaches.**

| Scoring task | Context | Manual | Semi-auto | $p$-value | Effect size |
|---|---|---|---|---|---|
| Sleep staging | TIB | 0.76 [0.69, 0.80] | 0.80 [0.76, 0.83] | < 0.0001* | -0.7146 |
| Leg Movements | TIB | 0.72 [0.64, 0.79] | 0.91 [0.86, 0.95] | < 0.0001* | -2.0223 |
| | Wake | 0.67 [0.57, 0.77] | 0.89 [0.82, 0.94] | < 0.0001* | -1.7121 |
| | Sleep | 0.75 [0.65, 0.81] | 0.92 [0.86, 0.95] | < 0.0001* | -1.6704 |
| Respiratory events | Apnea, Hypopnea, RERA (TIB) | 0.55 [0.43, 0.78] | 0.66 [0.53, 0.89] | < 0.0001* | -0.8315 |
| | Apneas (TIB) | 0.74 [0.35, 0.88] | 0.88 [0.57, 0.98] | < 0.0001* | -0.3783 |
| | Hypopneas (TIB) | 0.46 [0.36, 0.53] | 0.61 [0.51, 0.68] | < 0.0001* | -0.9569 |
| EEG Arousals | TIB | 0.58 [0.48, 0.65] | 0.65 [0.56, 0.71] | < 0.0001* | -0.6166 |

n = 330 resulting from all distinct combinations of expert scorer pairs (n = 66) on each of the five recordings involved in the corresponding scoring task. Distributions are characterized using the corresponding median and interquartile ranges. TIB = Time in Bed.

*Statistically significant result.

For the latter, higher agreement is obtained during sleep periods than during wakefulness. With regard to the analysis of respiratory activity, and attending to the different event sub-types, higher agreement is achieved for the scoring of apneas than of hypopneas. Finally, reliability associated with the scoring of EEG arousals reaches agreement levels similar to those obtained for the identification of respiratory events in general (i.e. apneas, hypopneas and RERAs altogether).

Individualized per-recording analyses for each of the tasks are supplied in Tables B1-B8 in S2 Appendix.

## Analysis of derived diagnostic indices

Table 4 examines inter-scorer agreement among the selected list of diagnostic parameters for the manual and semi-automatic scoring approaches. Agreement is evaluated using the Inter-class Correlation Coefficient (ICC).

When comparing absolute ICC values among the different tasks, a trend can be seen toward higher inter-scorer agreement when using the semi-automatic scoring approach, with the only exception of SOL. Regardless of the scoring approach, the highest absolute values of agreement are achieved for indices SE, AI, and WASO (ICC > 0.99 in all cases). Agreement associated with the scoring of apneas probably contributes to the relative high scores achieved for the AHI too. Detection of hypopneas, as reflected by HI on the other hand, shows relative lower levels of ICC agreement. HI is, in fact, is the index where the lowest overall agreement is achieved, followed by ArI. For all the examined indices, and regardless of the scoring approach, the obtained values represent significant scores when no a priori agreement is assumed in the null hypothesis ($p < 0.0001$ for all indices when $r0 = 0$).

For examining statistical significance of the observed differences between the manual and semi-automatic approaches, the null hypothesis is set to match baseline ICC levels obtained during manual scoring (column $r0$ in Table 3). In this case, significant differences are obtained for the indices of WASO, LMI, PLMI, AI and ODI. For SE, AHI, HI and ArI, the trend remains consistent toward higher ICC values when using the semi-automatic scoring

**Table 4. Comparison of inter-scoring agreement among diagnostic indices between manual and semi-automatic approaches.**

| Index (TST) | Summary of index distributions | | ICC | | $r0$ | ICC $p$-value |
|---|---|---|---|---|---|---|
| | **Manual** | **Semi-auto** | **Manual** | **Semi-auto** | | |
| SE (%) | 89.11 [68.05, 94.87] | 90.31 [70.59, 94.86] | 0.99 (0.98–1.00) | 1.00 (0.99–1.00) | 0.9938 | 0.0665 |
| SOL (min) | 5.27 2.50, 107.41] | 4.11 [2.31, 104.81] | 0.87 (0.69–0.98) | 0.84 (0.62–0.98) | 0.8720 | 0.5589 |
| WASO (min) | 86.20 [22.27, 151.77] | 79.18 [23.02, 139.51] | 0.99 (0.98–1.00) | 1.00 (0.99–1.00) | 0.9924 | 0.0481* |
| LMI | 25.27 [19.78, 70.19] | 31.73 [20.89, 61.82] | 0.92 (0.79–0.99) | 0.98 (0.93–1.00) | 0.9227 | 0.0166* |
| PLMI | 13.91 [9.20, 61.70] | 14.14 [12.79, 53.70] | 0.94 (0.82–0.99) | 0.98 (0.93–1.00) | 0.9351 | 0.0282* |
| AHI | 5.86 [3.71, 12.47] | 6.54 [4.70, 14.61] | 0.99 (0.96–1.00) | 0.99 (0.98–1.00) | 0.9878 | 0.1901 |
| AI | 1.26 [0.31, 3.46] | 1.57 [0.00, 3.54] | 1.00 (0.99–1.00) | 1.00 (1.00–1.00) | 0.9962 | 0.0441* |
| HI | 3.10 [2.38, 4.87] | 4.55 [3.13, 6.68] | 0.60 (0.31–0.93) | 0.75 (0.48–0.96) | 0.6010 | 0.1160 |
| ODI | 6.48 [4.24, 14.38] | 12.04 [6.01, 15.89] | 0.84 (0.62–0.98) | 0.98 (0.95–1.00) | 0.8370 | <0.0001* |
| ArI | 17.57 [12.98, 25.64] | 18.38 [13.82, 25.41] | 0.68 (0.39–0.95) | 0.77 (0.48–0.97) | 0.6824 | 0.2290 |

n = 60 resulting from all twelve scoring experts and the five analyzed recordings for each of the corresponding tasks. Summary index distributions are characterized using the respective median and interquartile ranges. Agreement is characterized in terms of Interclass Correlation Coefficient (ICC) with 95% confidence intervals. r0 = null hypothesis for baseline ICC score; SE = Sleep Efficiency; SOL = Sleep Onset Latency; WASO = Wake After Sleep Onset; (P)LMI = (Periodic) Leg Movement Index; AHI = Apnea-Hypopnea Index; AI = Apnea Index; HI = Hypopnea Index; ODI = Oxygen Desaturation Index; ArI = Arousal Index.
*Statistically significant result.

approach, albeit analyses do not reach statistical relevance. Only for SOL higher ICC values are obtained under manual scoring, nevertheless not reaching the level of statistical significance.

Individualized per-recording analyses are supplied in Tables C1-C10 in S3 Appendix. In this case manual vs. semi-automatic differences are examined both using paired Wilcoxon sign-rank and unpaired Brown-Forsythe tests, as described in the methods section.

## Discussion

The main goal of this study was to evaluate the possible benefits of using semi-automatic scoring of PSGs in comparison to classical manual visual approach. For this purpose, we have individually considered four of the most common subtasks involved in the analysis of PSGs: sleep staging, scoring of leg movements, detection of respiratory events, and of EEG arousals. On each case, quantifiable metrics of performance regarding the scoring time, and inter-scorer agreement, have been examined and compared among the two methods. To our knowledge, this is the first study to systematically address the differences between manual and semi-automatic scoring.

Our experimentation has shown that the use of semi-automatic analysis has benefits in the form of faster scoring and higher inter-scorer agreement. Faster scoring can help lowering down the associated diagnostic costs, and have a contribution toward reducing waiting lists as a consequence of the more efficient scoring production rate. Higher inter-scorer agreement translates to better consistency and reliability of the PSG outcomes, and therefore improved quality of the diagnosis. The trend is consistent across all the four examined tasks. Differences between the two approaches have achieved statistical significance both for the scoring time and the expert agreement. The impact of these differences on a subset of derived diagnostic indices, analyzed in terms of ICC agreement, has shown a more heterogeneous pattern. While statistical significant differences have been observed for indices of WASO, LMI, LMI, AI and ODI, statistical significance was not reached for indices of SE, SOL, AHI, HI and ArI. Still, for all the examined indices with the exception of SOL, the trend was consistent toward higher ICC values when using the semi-automatic scoring approach.

Structured and more detailed analysis of the main findings of this study in the context of the state-of-the-art is provided in the following subsections.

### Scoring time

To the authors knowledge this is the first study reporting and comparing the time associated with the scoring of respiratory events (both for the manual and semi-automatic approaches). Our data shows a median gain factor of 1.63 when using semi-automatic scoring. That we know of, and excluding preliminary estimations from our own group [32,39], this is also the first study to report on time associated to manual and semi-automatic detection of leg movements and EEG arousals. Specifically, a 2.41 gain factor (44.53 minutes for the manual approach, and 18.50 minutes when using the semi-automatic procedure) for the leg movements' detection task was obtained. This is similar to the reference previously reported in Roessen et al. [39], who nevertheless used an older version of the associated clinical scoring guidelines. With regard to the scoring of EEG arousals, a gain factor of 1.26 (median of 27.50 minutes for manual vs. 21.78 for semi-automatic) was obtained in the present study, not far from results reported previously in [32], and using the same clinical reference and automatic scoring algorithm, but on a different selection of PSG recordings. Last, with respect to sleep staging, some works can be found that have already examined the associated scoring times [10,22,23]. Anderer et al. [10], for example, have reported an average improvement from 84 to 5 minutes when using semi-automatic scoring, resulting in a gain factor of 5, well above the

results reported in this study. This could be explained by quality control mechanism implemented in their approach [40], considerably reducing the number of epochs subject to human supervision (on average, only 4% of epochs were changed by the 2 experts involved in [10]). Koupparis et al. [22], on the other hand, have reported an average 3 hours for manual scoring baseline, which could be improved to 45 minutes with the use of semi-automatic scoring. Younes et al. [23] have shown differences between full and minimal human intervention associated to semi-automatic sleep staging involving 50 and 6 minutes, respectively, on average. Baseline time for manual scoring, however, was not reported in their study. Our data has shown that 1.33 gain factor in scoring time associated to sleep staging when using semi-automatic scoring, in addition, while improving median inter-scorer agreement from K = 0.76 to K = 0.80. Even if the full-review semi-automatic sleep staging approach followed in this study might be suboptimal in terms of possible time gains (e.g. in comparison to partial quality-check guided as in Anderer et al. [10]), our data suggest that full-review of automatic sleep staging can still redound in cost savings regardless of what suggested in Svetnik et al. [20]. A general warning when comparing results from different works in the literature, is that one has always to bear in mind that relevant differences might exist between the respective population samples, the analysis methods, or the clinical scoring references valid at the time of the study.

## Inter-rater agreement: Kappa scores

There is abundant literature on the analysis of inter-scorer reliability of event markings in the scoring of sleep studies. Nevertheless, here most of available data regard almost exclusively to the manual scoring of sleep stages [4–6,13,41–49], with only few examples examining the case for other scoring tasks [5,50,51]. That we know of, this is the first study to provide results on the analysis of inter-scorer reliability under the context of semi-automatic scoring for the detection of leg movements, respiratory events, and EEG arousals.

A recent publication by the authors included a review of the related literature on manual sleep staging, resulting in kappa coefficients ranging widely between = 0.46–0.89 [26]. This range is compatible with the median agreement achieved in this study ( = 0. 76) for the manual scoring task. Inter-scorer agreement of sleep staging in the context of semi-automatic analysis has been examined previously by Anderer et al. [10], who reported an increase from = 0.76 to 0.99 when manual hypnograms were rescored semi-automatically. Similarly, Younes et al. [23], obtained an increase in associated paired kappa scores between two scorers from 0.71 to 0.95 with the help of a third scorer. In Koupparis et al., on the other hand, inter-scorer agreement reached a maximum average of = 0.61 with semi-automated analysis of the so-called hypnospectrogram [22]. In the same line, in Svetnik et al. [20] epoch-by-epoch agreement between scorers performing full or partial review of automated scoring ranged = 0.60–0.63. Reference agreement baseline for manual scoring, however, was not reported in the works of Koupparis et al. and Svetnik et al. In our study we have reported an increase from = 0.76 to = 0.80 with the use of semi-automatic scoring, which is a lower gain than in the works of Anderer et al. and Younes et al. (up to = 0.95–0.99). The higher agreement in these works can be regarded to the use of screening quality-management mechanisms and/or computer-derived features that significantly reduce the number of epochs subject to manual rescoring [40,52].

Only one past reference was found examining inter-scorer manual agreement in the detection of leg movements. In the study of Pittman et al. [5] = 0.77 was obtained between two scoring experts on a dataset of 31 PSGs. Notice, however, that agreement reported in Pittman et al. refers only to the scoring of PLMs, not LMs, and that the scoring reference was based on older standards (ASDA1993 [53]). Moreover, analysis was constrained to sleep periods only, and its

associated resolution was 30s. Our results, involving 12 scorers, and using the more recent WASM2016 scoring standards, resulted in global $\kappa$ = 0.72 for manual, and $\kappa$ = 0.91 for semi-automatic scoring, when examining LMs during TIB, and using a 0.5s analysis step. Agreement falls respectively to $\kappa$ = 0.67 and $\kappa$ = 0.89 during wake periods, and improves to $\kappa$ = 0.75 and $\kappa$ = 0.92 during TST.

Pitman et al. have reported as well a $\kappa$ = 0.82 for the manual scoring of apneas and hypopneas using the 2001 AASM Medicare scoring definitions on a 30s analysis epoch [5]. With our settings, we have achieved rather lower agreement resulting in median $\kappa$ = 0.55 (improving to $\kappa$ = 0.66 with semi-automatic scoring). We have obtained higher agreement for the scoring of apneas (median $\kappa$ = 0.74 for manual, $\kappa$ = 0.88 for semi-automatic) than for the case of hypopneas (respectively $\kappa$ = 0.46 and $\kappa$ = 0.61). This is an expected result, however no study that we know of had attempted to quantify this difference in terms of kappa agreement so far.

As for the EEG arousal task, some studies can be found reporting kappa values for manual scoring in the $\kappa$ = 0.47–0.59 range [32,50,51]. Once again, some of these studies use older scoring guidelines (ASDA1992) besides other sources of variability, and therefore direct comparison has to be carefully considered. Regardless, the reported range is consistent with our experimental results in the case of manual scoring ($\kappa$ = 0.58). Our study shows, in addition, that better inter-scorer agreement can be achieved if using semi-automatic scoring (up to $\kappa$ = 0.65 in our dataset).

## Inter-rater agreement: Diagnostic indices

We have found three preceding works that examined differences between manual and semi-automatic scoring related to diagnostic indices reported in our study. In Svetnik et al. no significant differences between the two approaches were found in the resulting indices for SOL and WASO [20]. This result matches our trend in the case of SOL, but not for WASO. Koupparis et al., on the other hand, have reported ICC values for WASO of 0.91 under full-editing semi-automatic review, considerably decreasing to ICC of 0.05 under a minimal editing approach [22]. In the same work, a similar trend was reported for SE. Punjabi et al. [54], instead, have found no significant differences in related calculations of SE, which more closely matches the outcome of our experimentation. Likewise, our work agrees with the results reported in the work of Punjabi et al., who found no relevant differences among corresponding ICC scores of AHI and ArI between manual and semi-automatic scoring. We were not able to find any other references in the literature for the remaining indices examined in our study in relation to semi-automatic scoring.

In the context of manual analysis, on the other hand, one can find several other past studies reporting on inter-rater related ICC agreement scores [5–7,54–60]. The specific values of agreement vary per study. Danker-Hopfe et al. [6] and Kuna et al. [60], for example, have reported ICC values for SE of 0.91 and 0.77 respectively, which is below the agreement obtained in this study (ICC = 0.99). Reliability on PLMI has been reported by Pittman et al. [5] (ICC = 0.93) and Bliwise et al. [55] (ICC = 0.91–0.99), however, using older definitions of the index [61,62]. This is relevant as recent studies [63–65] have pointed out to significant differences in the resulting PLMI calculations when using as reference the latest clinical scoring guidelines. The agreement results are nevertheless comparable to the levels obtained in our work (ICC = 0.94), which use the recent WASM2016 standards [3]. Under this reference, our study is in fact the first one to set a reference for the inter-expert agreement associated with the LM and PLM indices (ICC = 0.92 and 0.94, respectively). With regard to PSG respiratory-derived indices, possibly the most widely reported is the AHI, with reliability scores for manual scoring ranging widely between ICC 0.54–0.99, depending on the consulted study

[5,7,11,54,60]. Most likely, these differences are to a great extent driven by the specific rule used for the scoring of hypopneas. As stated before, it is widely accepted that agreement regarding scoring of hypopneas is lower as compared to that of apneas. This can also be observed by comparing ICC agreement values associated with AI and HI indices in the referenced literature. This is also the case in our study, with ICC agreement for the manual derivation of respiratory related indices following the expected trend of higher agreement for AI (ICC = 0.99) in comparison to HI (ICC = 0.60). Finally, reliability reports of ArI, related to manual scoring of EEG arousals, show even more variability across the existing literature (ICC = 0.09–0.96 [5,41,54–58]). Our results fit approximately in the middle of that range (ICC = 0.68) improving to ICC = 0.76 when using semi-automatic scoring.

## Limitations and concluding remarks

Some limitations of our study have to be mentioned. First, it is important to remark that absolute values of the various investigated performance scores are associated with one specific sleep lab. This study does not involve analysis of inter-scorer variability across multiple centers, and thus results might not generalize to other centers. In such scenario, the respective values of scoring agreement are expected to be lower in comparison due to the greater amount of variability involved [2,60]. This study neither has attempted to quantify the corresponding levels of intra-scorer variability within any of the two examined approaches (manual or semi-automatic). Thus, it cannot completely be ruled-out that some of the differences between manual and semi-automatic approaches could be influenced by a component of intra-scorer variability effect, at the individual scorer level at least. Nevertheless, the relative high number of involved experts (12 in our study) should contribute to limit its the impact on the global results. In addition, although a 4-month separation between manual and semi-automatic rescoring could be regarded as a safe margin in practice, randomized order would have probably been a better choice from a methodological point of view.

It should also be remarked that quality indicators derived from the semi-automatic scoring procedure are likely modulated by the reliability and performance of the specific automatic analysis algorithms used in the first instance. One might speculate with the idea that the better the algorithm, the higher the improvement on expert agreement with respect to the manual approach. However, there is no actual evidence that allows us to support this hypothesis. The usage of alternative automatic scoring methods might lead to different results. Regardless, our results support the hypothesis that semi-automatic algorithm can improve scoring quality in terms of both speed and resulting inter-scorer agreement. Also interesting, inter-scorer reliability studies available through literature, and this is no exception, implicitly assume that the outcome of all human scorers is equally valid. This might be a risky assumption, although there is no clear formula to discern who (out of a set of human experts) represents the best reference, and who does not. This propounds an interesting line of future research linked to another non-less interesting debate: can (full) automatic scoring outperform human experts? Of course, in terms of its capacity to correctly identify the relevant events associated with the physiological activity's ground truth. There is no debate that automatic analysis can outperform manual scoring in terms of speed (and our study has shown this is also possible under a semi-automatic context). If, like in this case, the standard reference is subject to variability associated with human decision, it does not seem very plausible that any automatic algorithm could perform beyond the limit set by the average human agreement. After all, as stated before, deviations from such a reference do not necessarily correlate with the quality of the associated scorings. This is a subject that deserves more study.

Last but not least, one another important limitation of this study relates to the number of PSGs involved in the evaluation of each analysis subtask. The relative high number of sleep experts involved (12 for our study) partially counteracts this fact, and indeed, the number has proven enough to reach statistical significance among many of the reported hypothesis tests. However, a higher number of PSGs per task would be in general desirable. More specifically, for those cases in which the reported trends did not achieve significant effects, the question remains open on whether this could be attributed to the relative small PSG sample size. Notice, on the other hand, that post-hoc power analyses were consciously omitted because no useful conclusions are expected from them [66]. A higher sample size would also contribute to spread the bias risk due to demographic and physiological subject variability. Unfortunately, the chosen sample size was imposed by the available resources; thus this was not a design parameter we were able to tune. As noticed, scoring of PSG data is complex and time-consuming, and expert's time is expensive and scant.

In conclusion, our results provide an updated reference for inter-scorer agreement levels and scoring times associated with both manual and semi-automatic scoring of PSG studies. We have systematically analyzed and compared the resulting differences, showing that the use of semi-automatic scoring can improve both speed and consistency of the PSG analysis outcomes. With a more efficient production rate diagnostic costs can be reduced and diagnostic times can be shortened. Enhancement of inter-scorer agreement, in addition, results in higher repeatability and quality of the diagnosis. More work has to be done to investigate generalization of these results by increasing the subject sample and its heterogeneity. Future work should also assess the effects of inter-center and intra-expert scoring variability, and goodness of fully automatic scoring in comparison to manual and semi-automatic approaches.

## Supporting information

**S1 Appendix. Scoring time individualized per-recording and per-scorer analyses.**
(DOCX)

**S2 Appendix. Kappa-agreement individualized per-recording analyses.**
(DOCX)

**S3 Appendix. Diagnostic-indices individualized inter-scorer variability per-recording analyses.**
(DOCX)

**S4 Appendix. Correlation analyses for automatic selection of PSG recordings involved in the study.**
(DOCX)

**S1 Dataset. Scoring EDF+ annotation files.**
(ZIP)

## Acknowledgments

Authors want to thank support received from Haaglanden Medisch Centrum Sleep Center management staff and the group of expert sleep technicians participating in the PSG rescoring task involved in this study for their fundamental effort and contribution.

## Author Contributions

**Conceptualization:** Diego Alvarez-Estevez.

**Data curation:** Roselyne M. Rijsman.

**Formal analysis:** Diego Alvarez-Estevez.

**Funding acquisition:** Diego Alvarez-Estevez.

**Investigation:** Diego Alvarez-Estevez.

**Methodology:** Diego Alvarez-Estevez.

**Project administration:** Diego Alvarez-Estevez.

**Resources:** Roselyne M. Rijsman.

**Software:** Diego Alvarez-Estevez.

**Supervision:** Roselyne M. Rijsman.

**Validation:** Diego Alvarez-Estevez.

**Visualization:** Diego Alvarez-Estevez.

**Writing – original draft:** Diego Alvarez-Estevez.

**Writing – review & editing:** Diego Alvarez-Estevez, Roselyne M. Rijsman.

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
