## [Decision Letter · Decision Letter 0]

14 Jul 2022

PONE-D-22-14573Computer-assisted analysis of polysomnographic recordings improves inter-scorer associated agreement and scoring timesPLOS ONE

Dear Dr. Alvarez-Estevez,

Thank you for submitting your manuscript to PLOS ONE. After careful consideration, we feel that it has merit but does not fully meet PLOS ONE’s publication criteria as it currently stands. Therefore, we invite you to submit a revised version of the manuscript that addresses the points raised during the review process. 

We look forward to receiving your revised manuscript.

Kind regards,

Christian Veauthier, M.D.

Academic Editor

PLOS ONE

Journal Requirements:

"This study has been initiated at Haaglanden Medisch Centrum under project number 2019-073. The study has also been partially funded under project ED431H 2020/10 of Xunta de Galicia. Authors wish to acknowledge the support received from the Centro de Investigación de Galicia (CITIC), funded by Xunta de Galicia and the European Union (European Regional Development Fund-Galicia 2014-2020 Program), by grant ED431G 2019/01"

Please include this amended Role of Funder statement in your cover letter; we will change the online submission form on your behalf

Reviewers' comments:

Reviewer's Responses to Questions

**Comments to the Author**

1. Is the manuscript technically sound, and do the data support the conclusions?

Reviewer #1: Partly

Reviewer #2: Yes

Reviewer #3: Partly

2. Has the statistical analysis been performed appropriately and rigorously? 

Reviewer #1: N/A

Reviewer #2: Yes

Reviewer #3: Yes

3. Have the authors made all data underlying the findings in their manuscript fully available?

Reviewer #1: No

Reviewer #2: Yes

Reviewer #3: Yes

4. Is the manuscript presented in an intelligible fashion and written in standard English?

Reviewer #1: Yes

Reviewer #2: Yes

Reviewer #3: Yes

5. Review Comments to the Author

Reviewer #1: The authors investigated inter-scorer agreement with 12 “expert technician scorers” on 5 PSG each selected for sleep staging, EEG arousals, “respiratory activity” and “identification of leg movements”. They found that when semi-automatic or computer assisted sleep scoring was used, the time for sleep analysis was reduced and suggest this could help to reduce waiting times for sleep laboratories.

I have several critical comments. First of all, the idea is not novel, and multiple methods have been described to assist sleep scoring, and this is only another of hundreds of papers on this topic. Second, from a methodological standpoint, it seems to lack a certain rigorousness, this begins with the abstract, and goes on within the text.

They talk about “sleep stating” (only other instances they correctly refer to sleep staging), then they state in the abstract that they want to detect “EEG arousals”, but later refer to ”micro-arousals” or arousals alone. The use of different terms for the same or different types of arous-als is confusing and needs to be defined.

The same is true for the “respiratory activity” (in the abstract), where they say respiratory events further down in the text, and even use the term “respiratory function”. If they just did done regular scoring of respiratory variables, they should call it as such and not as respiratory activity or respiratory function.

The same is the identification of leg movements: it is okay to do leg movements, but they also should do PLM indices.

Furthermore, the authors conclude that the use of computer assisted scoring could help to re-duce waiting lists of sleep laboratories: usually, the waiting lists of sleep laboratories are long not because technicians never finalize their scoring, but because of restrictions in polysomnog-raphy units and polysomnography beds.

As a general remark, while it seems nice that they selected PSG for sleep staging, respiratory, LM etc., in today’s advanced background knowledge on automatic scoring one would at least expect that they define larger samples from well-defined patient and diagnostic groups. The scoring algorithm should be made public, and not simply referred to dozens of own studies.

There were several statements made, which are not backed up by evidence, for instance: “semi-automatic scoring, therefore, effectively results in valid …”

Methods:

the authors used “PSG data for this study has been gathered by retrospective inspection…”, how can they make sure that this did not introduce a large bias? It is also unclear what they mean with statements like “no patients was therefore subjected to any additional behavior in relation to this study, nor was prescribed any additional treatment outside of the regular clinical workflow”, this is superfluous, as in a retrospective study no patient will undergo any treatment or “additional behavior”.

Line 84: in the present study a group of sleep technicians were prompted to review… did these sleep technicians have any type of certification background? Were they 12 technicians or other scorers?

Sleep scoring:

here, the authors present an exaggerated number of self-citations (eight), and even refer the reader “to check the corresponding references” (!). This is annoying, why the reader should bother to read the present manuscript, if he is referred to so many references to check?

What is the use of the present ms, only to report that scoring time can be reduced in a not fur-ther described small sample with respect to not further described scoring criteria and with a one of multiple algorithms?

It is furthermore unclear why they selected fivexfour PSG from 2801 recordings.

So finally they selected 20 PSG recordings, on the basis of what?

This could be biased bias, or did they simply use the first best available?

Did they screen the 2801 or do they just want to express the magnitude of the database?

In summary, this is a very long manuscript, which adds very little new knowledge.

Some statements are not based on data, for instance, faster scoring means … the possibility to reduce the waiting list by the consequent increase in the scoring production: usually the limited availability of PSG is based on the limited number of PSG places, not on scoring time.

Also, the authors, while on the one side extensively self-citing, do not cite others: for instance: with only few examples addressing the case of other scoring tasks: this is a very general term, and not backed by any reference.

As long as the authors do not make available their software, the results cannot be reproduced by others. One should keep in mind, that this is a single software, among hundreds of sleep scoring etc. softwares, and therefore the results cannot be generalized.

Reviewer #2: The authors analysed the possible benefits of semi-automatic PSG scoring of sleep staging, EEG arousals, respiratory events and limb movements, compared to manual visual approach. They provide quantitative metrics of performance regarding scoring time and inter-scorer agreement, showing the benefit deriving from semi-automatic scoring of the above mentioned PSG parameters and providing a quantification of the time saved in this way.

The manuscript is clearly written, data are extensively provided, results well and clearly presented, the discussion is complete and clear and limitations are acknowledged. I do not have any comment.

Reviewer #3: This paper presents very important work on evaluating the effect of autoscoring in PSG sleep analysis, a field that is still ridden with quite some skepticism toward automation. In principle, this paper should be published, however, some major weaknesses still need to be addressed:

- the authors do not seem to be fully aware of the state of the art, when judged by the references given in line 25. Systems that are actually used in the field are Morpheus (cited via Pittman et al.), Somnolyzer, Michelle and, most recently, Ensosleep/Ensodata (see also table 2 in Fiorillo et al.). Such systems deserve more attention when discussing the results of the authors' own algorithms. While it is true that time savings have not been published for any of those systems, Anderer et al., Neuropsychobiology 2010;62:250–264 did look into agreements between different sem-automatic scorings. The striking property of those results is the much higher agreement after correction than are achieved in this paper. This points to the fact that the biggest gains from autoscoring are obtained when scorers are NOT allowed to change results freely, but instead are trained to recognize where correction is necessary (e.g. because signal quality was bad). Applying the strategy that the authors have used in this paper instead seems to entail much more extensive changes than are necessary to achieve valid clinical results (pretty much like allowing one scorer to change the output of another expert scorer to their own liking) - see for instance, k=0.99 in Anderer et al., vs,. k=0.8 in table 3. Also Punjabi et al., Sleep 38(10), 2015, seems to point in a similar direction. Thus the results presented here might have an unfair bias against autoscoring, which should be mentioned in the discussion.

- the other weakness is the rather low number of recordings (5) used in each comparison. While this is addressed in the discussion, it still is very unfortunate. For instance, this means that each ICC is calculated based on 5 samples only, which is rather questionable. Why weren't all 4 tasks (staging, arousals, apneic events, leg movements) performed on each recording, which would not only increase the sample sizes but also much more correspond to the reality in a sleep lab, where these tasks are never solved in isolation? If those additional scorings can no longer be added to the study, then what remains is a much larger weakness due to sample size than currently acknowledged in the paper.

- on the other hand, some of the high ICC values (compare, e.g., again against Punjabi et al.) for some of the endpoints suggests that due to the fact that all scorers came from the same lab leads to an overestimation of some agreements, which again blures the advantages of autoscoring somewhat. This, too, needs more acknowledgement.

- Finally one must mention that the strategy to always let scorers perform visual scoring first and semi-automated scoring second, despite the 4 months of separation, introduces a bias. A randomized order would have been much preferred.

In summary, this is good and important work but hampered by considerable shortcomings that only partly can be overcome.

6. PLOS authors have the option to publish the peer review history of their article (what does this mean?). If published, this will include your full peer review and any attached files.

Reviewer #1: No

Reviewer #2: No

Reviewer #3: No

---

## [Author Response · Author response to Decision Letter 0]

13 Aug 2022

See attached "response to reviewers" document

---

## [Decision Letter · Decision Letter 1]

19 Sep 2022

Computer-assisted analysis of polysomnographic recordings improves inter-scorer associated agreement and scoring times

PONE-D-22-14573R1

Dear Dr. Alvarez-Estevez,

We’re pleased to inform you that your manuscript has been judged scientifically suitable for publication and will be formally accepted for publication once it meets all outstanding technical requirements.

Kind regards,

Christian Veauthier, M.D.

Academic Editor

PLOS ONE

Additional Editor Comments (optional):

Reviewers' comments:

Reviewer's Responses to Questions

**Comments to the Author**

1. If the authors have adequately addressed your comments raised in a previous round of review and you feel that this manuscript is now acceptable for publication, you may indicate that here to bypass the “Comments to the Author” section, enter your conflict of interest statement in the “Confidential to Editor” section, and submit your "Accept" recommendation.

Reviewer #2: All comments have been addressed

Reviewer #3: All comments have been addressed

2. Is the manuscript technically sound, and do the data support the conclusions?

Reviewer #2: Yes

Reviewer #3: Yes

3. Has the statistical analysis been performed appropriately and rigorously? 

Reviewer #2: Yes

Reviewer #3: Yes

4. Have the authors made all data underlying the findings in their manuscript fully available?

Reviewer #2: Yes

Reviewer #3: Yes

5. Is the manuscript presented in an intelligible fashion and written in standard English?

Reviewer #2: Yes

Reviewer #3: Yes

6. Review Comments to the Author

Reviewer #2: Comments have been addressed and the manuscript improved. I do not have any further comment.

Reviewer #3: (No Response)

7. PLOS authors have the option to publish the peer review history of their article (what does this mean?). If published, this will include your full peer review and any attached files.

Reviewer #2: No

Reviewer #3: No

---

## [Editor Report · Acceptance letter]

21 Sep 2022

PONE-D-22-14573R1 

Computer-assisted analysis of polysomnographic recordings improves inter-scorer associated agreement and scoring times 

Dear Dr. Alvarez-Estevez:

I'm pleased to inform you that your manuscript has been deemed suitable for publication in PLOS ONE. Congratulations! Your manuscript is now with our production department. 

Kind regards, 

on behalf of

Dr. Christian Veauthier 

Academic Editor

PLOS ONE